# Transmission Dynamics, Heterogeneity and Controllability of SARS-CoV-2: A Rural–Urban Comparison

**DOI:** 10.3390/ijerph18105221

**Published:** 2021-05-14

**Authors:** Yuying Li, Taojun Hu, Xin Gai, Yunjun Zhang, Xiaohua Zhou

**Affiliations:** 1Department of Biostatistics, School of Public Health, Peking University, Beijing 100191, China; liyuying@pku.edu.cn (Y.L.); 2011110158@stu.pku.edu.cn (T.H.); gaitianmu@outlook.com (X.G.); yunjun.zhang@pku.edu.cn (Y.Z.); 2Beijing International Center for Mathematical Research, Peking University, Beijing 100871, China; 3Center for Statistical Sciences, Peking University, Beijing 100871, China

**Keywords:** SARS-CoV-2, urban–rural, heterogeneity, vaccination, non-pharmaceutical interventions

## Abstract

Few studies have examined the transmission dynamics of severe acute respiratory syndrome coronavirus 2 (SARS-CoV-2) in rural areas and clarified rural–urban differences. Moreover, the effectiveness of non-pharmaceutical interventions (NPIs) relative to vaccination in rural areas is uncertain. We addressed this knowledge gap through using an improved statistical stochastic method based on the Galton–Watson branching process, considering both symptomatic and asymptomatic cases. Data included 1136 SARS-2-CoV infections of the rural outbreak in Hebei, China, and 135 infections of the urban outbreak in Tianjin, China. We reconstructed SARS-CoV-2 transmission chains and analyzed the effectiveness of vaccination and NPIs by simulation studies. The transmission of SARS-CoV-2 showed strong heterogeneity in urban and rural areas, with the dispersion parameters *k* = 0.14 and 0.35, respectively (*k* < 1 indicating strong heterogeneity). Although age group and contact-type distributions significantly differed between urban and rural areas, the average reproductive number (*R*) and *k* did not. Further, simulation results based on pre-control parameters (*R* = 0.81, *k* = 0.27) showed that in the vaccination scenario (80% efficacy and 55% coverage), the cumulative secondary infections will be reduced by more than half; however, NPIs are more effective than vaccinating 65% of the population. These findings could inform government policies regarding vaccination and NPIs in rural and urban areas.

## 1. Introduction

The dynamics of an outbreak depend on the average reproductive number (*R*) and individual heterogeneity in transmission. Although there is ample research on R, studies regarding heterogeneity are limited. Heterogeneity reflects the divergence of secondary infections in each case and can be estimated by describing the distribution of secondary cases as a negative binomial distribution with dispersion parameter k, where *k* < 1 suggests that the transmission is over-dispersed [1]. Diseases with high heterogeneity show infrequent but explosive epidemics; for example, in 2003, many settings experienced no epidemic despite unprotected exposure to severe acute respiratory syndrome (SARS) cases [2,3], whereas a few cities suffered explosive outbreaks of SARS [4,5]. Thus, understanding the role of transmission heterogeneity in severe acute respiratory syndrome coronavirus 2 (SARS-CoV-2) dynamics is important for outbreak control.

However, few studies explore the impact of asymptomatic infection on disease dynamics, especially individual heterogeneity in transmission. This is likely due to the lack of a valid statistical model and ambiguity of fundamental epidemiological questions that remain poorly understood, such as the proportion of asymptomatic cases [6]. Nishiura et al. estimated that the asymptomatic ratio of coronavirus disease (COVID-19) was 41.6% (5 out of 12 confirmed cases) among 565 Japanese individuals evacuated from Wuhan, China [7]. A recent review [8] of 41 studies showed that the pooled percentage of asymptomatic infection was 15.6% (95% confidence interval (CI): 10.1–23.0%).

Moreover, most studies focus solely on urban areas and ignore rural regions. As far as we know, Asghar et al. [9] found that the transmission of SARS-CoV-2 showed strong heterogeneity in rural United Kingdom with *k* = 0.35 (95% CI: 0.41–0.29). A study in Georgia [10] characterized age-specific infectiousness of SARS-CoV-2 transmission in four city areas and one rural area and showed strong heterogeneity, but lack of comparison. Rural regions tend to have higher levels of poverty [11] and fewer job opportunities [12] relative to urban areas. Furthermore, rural areas broadly lack access to healthcare [13], tend to have older and healthier populations [14,15], and lack awareness of timely medical treatment [16]. The gaps lead to discrepancy in transmission dynamics between urban–rural areas, warranting improved corresponding control policies.

The second-wave outbreak in Hebei, China, mainly occurred in rural regions and swiftly subdued. After the first confirmation on 2 January 2021, the government implemented city-wide nucleic acid tests (NAT) [17] in the two most severely affected cities to detect symptomatic and asymptomatic infections. The urban outbreak in Tianjin, a municipality and a coastal metropolis in China, occurred from 21 January 2020. Concurrently, highly detailed epidemiological information on individuals and their close contacts was collected by the Health Commissions of Hebei and Tianjin.

In this study, we characterized the difference of transmission dynamics and heterogeneity of the SARS-CoV-2 outbreak in rural and urban areas, arguing that the government should pay more attention to older adults, children, and community contacts when conducting prevention and control measures in rural areas. Additionally, we extend a statistical model that can be applied to other regions and provide more comprehensive results considering symptomatic and asymptomatic cases. To the best of our knowledge, although there are many studies modelling vaccination and non-pharmaceutical interventions (NPIs) [18,19], this is the first direct comparison of the effects of NPIs under actual conditions and vaccination.

## 2. Materials and Methods

### 2.1. Data Collection

We collect detailed data on 942 confirmed SARS-CoV-2 infections in Hebei Province, China, from 2 January to 20 February 2021, which are available in the website of the Health Commission of Hebei province [20]. In addition, asymptomatic infections accounted for 17% (194/1136) of the total infections. No new infections have been confirmed in Hebei province since February 14, indicating that the outbreak has been controlled. We also collect detailed data on 135 infections confirmed in Tianjin [21] from 21 January 2020 to 26 February 2020. The definition of rural and urban areas in China are established according to relative regulations (18). Primary and secondary SARS-CoV-2 infections are identified through (i) active screening of incoming passengers in Hebei Province, especially those have travelled to areas defined by the Chinese government [22] as medium- or high-risk to capture travel-associated symptomatic and asymptomatic infections; (ii) passive surveillance in hospitals and outpatient clinics, involving testing of individuals suspected to have COVID-19 in order to capture symptomatic cases; (iii) contact tracing of all confirmed infections identified by the above screening, followed by systematic monitoring of their close contacts, in order to capture symptomatic and asymptomatic infections; and (iv) city-wide NAT to capture symptomatic and asymptomatic infections. From 6 January to 22 January 2021, Shijiazhuang City, the most severely affected area in Hebei Province, has carried out three rounds of full-staff NAT, with the total exceeding 30 million person-times.

The collected data for each confirmed case included age, sex, prefecture, date of symptom onset, date of diagnosis of asymptomatic infection, date of confirmation, potential exposures, and contact history. On the basis of contact tracing, a SARS-CoV-2 cluster is defined as a group of ≥2 confirmed SARS-CoV-2 cases or asymptomatic infections with an epidemiological link [23], i.e., occurring through the same contact type (e.g., home, social, community, or other) [24]. According to the extent of resolution of the reconstructed infection cluster, i.e., the number of primary cases and chain size (total number of cases in the transmission chain), three chain types are further identified (simple/ordinary/complex). We consider the sporadic cases [25] as isolated simple transmission chains (detailed definitions are provided in Appendix A). Because contact tracing of asymptomatic infections is unavailable, we imputed them into the whole transmission chain according to the rates of secondary cases (imputation mechanisms are provided in Appendix A).

### 2.2. Statistical Analysis

#### 2.2.1. Inference about Transmission Characteristics

To estimate the *R* and *k* simultaneously, we deployed a statistical stochastic method on the basis of the Galton–Watson branching process [26] to simulate the entire transmission. Assuming that each case has the same probability of being asymptomatic with a discounted average reproductive number αR and the offspring distribution follows a negative binomial distribution with mean *R* and dispersion parameter *k* (lower *k* indicates higher heterogeneity), we estimated two parameters using maximum likelihood estimation (MLE) [27,28] and improved their method to fit into a wider population including symptomatic and asymptomatic cases. To guarantee the robustness of the estimation results, we adopted two approaches to infer the corresponding CIs of *R* and *k*, namely, the likelihood radio test (LRT) and biased-corrected and accelerated bootstrap (BCa bootstrap). The model details and parameter settings are provided in Appendix A.

#### 2.2.2. Estimation of Serial Interval

On the basis of the identified transmission chains and dates of symptom onset, we obtained 8 and 12 infector–infectee pairs [29] in rural and urban areas. To compare the difference between the serial intervals of SARS-CoV-2 in rural and urban areas, we fitted serial intervals retrieved into a parametric Weibull model. Statistically significant differences were determined by conducting the LRT.

#### 2.2.3. Assessment of Different Interventions

We analyzed the effectiveness of two types of interventions on SARS-CoV-2 transmission on the basis of parameters in rural areas: vaccination and NPIs, including city-wide NATs, isolation, and mask-wearing. Due to the large number of cases in Hebei and the inclusion of asymptomatic infections, it is better to evaluate the effects of different interventions under parameters of rural areas. For the NPIs, we divided the entire outbreak period into three segments according to the time of two rounds of city-wide NAT (9 January and 14 January 2021) and estimated *R* and *k* for each time period. We used the parameters before the first NAT intervention as the baseline and the parameters for the second period (during 9 January and 14 January 2020) as the effect of non-pharmaceutical intervention. To clarify the impact of the vaccine and its comparison with NPIs, we assumed that the population had been vaccinated before the outbreak (80% efficacy [30]) and simulated the cumulative secondary infections, considering a range of coverage rates (20%, 55%, 65%, and 75%). All interventions were compared with the baseline (without control measures).

## 3. Results

### 3.1. Characterizing SARS-CoV-2 Transmission Chains in Rural and Urban Areas

In total, 942 confirmed SARS-CoV-2 infections occurred in the rural outbreak, except for two infections with missing information, occurring in 387 (41.1%) males and 553 (58.8%) females. We observed the largest cluster involving 44 SARS-CoV-2–infected individuals. The 942 SARS-CoV-2-confirmed infections were grouped into 655 transmission chains, including 639 simple chains (752 infections, average size: 1.2), 15 ordinary chains (186 infections, average size: 12.4), and one complex chain (4 infections, average size: 4; Appendix A). Exposures were grouped into four categories according to contact type: household, community, social, and primary case (definitions in Appendix A). Except for the primary case, household contacts account for the highest proportion of transmission, followed by community contacts and social contacts. Figure 1A shows the reconstructed transmission chains in the rural area.

For the outbreak in the urban area, we confirmed 135 SARS-CoV-2 infections, including 72 (53.3%) males and 63 (46.7%) females. The largest cluster involved 45 SARS-CoV-2–infected individuals. The 135 infections were grouped into 43 transmission chains including 36 simple chains (47 infections, average size: 1.3), 5 ordinary chains (78 infections, average size: 15.6), and 2 complex chains (10 infections, average size 5; Appendix A). In the urban outbreak, household contacts also accounted for the highest proportion of transmission (Figure 1B).

### 3.2. Comparison of SARS-CoV-2 Transmission Characteristics between the Rural and Urban Areas

There are no significant differences in *R* and *k* between the two areas (*p* < 0.05, Table 1). Considering asymptomatic infections, the average *R* were 0.55 in the rural outbreak and 0.74 in the urban outbreak, which represented low transmission risks. The 95% CIs of *R* estimated by LRT and BCa bootstrap methods were not significant different, indicating the robustness of our results. The dispersion parameter *k* was 0.14 (95% CIs estimated by LRT and BCa bootstrap: 0.10–0.20 and 0.10–0.19) in rural outbreak and 0.35 (95% CIs estimated by LRT and BCa bootstrap: 0.13–1.21 and 0.12–0.95), indicating considerable heterogeneity of SARS-CoV-2 transmission in rural and urban areas. However, there were significant differences in age, sex, and contact-type distributions between these two areas. The proportions of older cases (≥65 years old) and child cases (≤20 years old) in the rural area were higher than those in the urban area (16.1% vs. 14.8% and 16.4% vs. 3.7%, respectively). Although more than 50% of transmissions in both urban and rural areas were caused by household contacts (61.1% and 51.4%), community contacts also accounted for a large proportion (46.5%) in rural areas. The median serial interval of the rural area was shorter than that of the urban area (5.5 days vs. 6.0 days), although without significant differences.

### 3.3. Assessment of NPIs and Vaccination in SARS-CoV-2 Transmission

For NPIs, until the first NAT, the average *R* was 0.81 (95% CI: 0.65–1.02) and dispersion parameter *k* was 0.27 (95% CI: 0.14–0.56; Table 2). After the first NAT and before the second NAT, *R* decreased significantly to 0.33 (95% CI: 0.22–0.50) and remained around this level after the second NAT, while the *k* value showed a downward trend at first but then increased slightly to 0.17 (95% CI: 0.10–0.31) without significant changes. We concluded that the first round of NAT played the most significant role in curbing the spread of SARS-CoV-2.

We conducted simulation studies with 656 primary cases and estimated the cumulative offspring infections from 1 January to 31 March. In Figure 2A, the cumulative offspring infections in the rural area are shown to be projected to reach 1575 (95% CI: 1110–2094) under the baseline setting (*R* = 0.81, *k* = 0.27). In other words, the total number of infections of the outbreak in the rural outbreak will reach 2231, as the number of primary cases was 656. Comparing the final secondary size in vaccinating different proportions of the population, with 55% of the population vaccinated, the cumulative offspring infections were found to reduce by 51% at 769 (95% CI: 468–1156). At 75% of the population vaccinated, the cumulative offspring infections would reduce by 70% at 478 (95% CI: 272–789). As shown in Figure 2B, the final secondary infections were found to be 264 (95% CI: 186–353), showing that the number of infected people would reduce by approximately 83%. Considering the effectiveness of interventions, we found that the effect of vaccinating all personnel would be better than that of the current NPIs implemented in rural areas, while current control is superior to vaccinating 65% of the population.

### 3.4. Sensitivity Analyses

In addition, the robustness of the model was verified. At first, we identified the role of asymptomatic infections in the epidemic dynamics. Given the higher proportion of asymptomatic infections, the estimates for *R* showed a steadily increasing trend, from 0.51 to 1.95. The estimates of *k* fluctuated at around 0.14. This indicates that with a higher estimation of the proportion of asymptomatic cases, on the basis of a parallel dataset, we found that the estimated *R* was larger, indicating a more severe potential spread of SARS-CoV-2. In contrast, there was no significant change in the heterogeneity of disease transmission due to the proportion of asymptomatic infections. Moreover, we evaluated the performance of our model on the basis of the idea of 10-fold cross-validation [31]. The data were divided into 10 equal parts, and nine of them were randomly selected for parameter estimation. The mean values (standard deviations) of *R* and *k* were 0.55 (0.02) and 0.14 (0.01), respectively, verifying the good performance of our model when generalized to an independent dataset (Table 3).

## 4. Discussion

We characterized the transmissibility of SARS-CoV-2 in rural and urban areas with the presence of asymptomatic infections on the basis of the detailed epidemiology records in Hebei and Tianjin, China, and further compared the effectiveness of vaccination and that of NPIs. SARS-CoV-2 transmission in the urban and rural areas showed a strong heterogeneity. Moreover, household contact was the most important mode of transmission, whether in the city or the countryside, but community contact also played an important role in countryside transmission. We also found that in the vaccination scenario (80% efficacy and 55% coverage), the cumulative secondary infections will be reduced by more than half; however, NPIs are more effective than vaccinating 65% of the population. The presence of asymptomatic infections might affect the estimation of *R* but showed no significant effect on estimating transmission heterogeneity.

The estimated dispersion parameter *k* was 0.14 (95% CI: 0.10–0.20) in the rural area, indicating strong transmission heterogeneity. This result was consistent with that of another study. Lau et al. [10] reported that a rural area (Dougherty) in Georgia, USA, had strong transmission heterogeneity (*k* = 0.43; 95% CI: 0.39–0.47). Although there was no significant urban–rural difference found in *k* under 5% type I error (*p* = 0.09), *k* in the rural area was lower than that in the urban area, in concordance with the results of Lau et al. [10]. Transmission heterogeneity results from many factors including pathogen virulence, control measures, and activity density. The results of whole-genome sequencing and phylogenetic analysis revealed that the strains of Tianjin and Hebei both belonged to the European branch of the lineage (L-Lineage) [32,33]. However, large-scale NAT, which was not carried out in Tianjin, can quickly and comprehensively screen asymptomatic and mild symptomatic infections, thereby effectively shortening the infection time. Additionally, the rural outbreak in Hebei coincided with China’s spring festival and several weddings, thus increasing the probability of large gatherings. Therefore, the government should pay sufficient attention to rural areas instead of focusing solely on urban areas.

SARS-CoV-2 transmission in rural areas have similar transmission dynamics as that in urban areas but differ in terms of age group and contact-type distributions. Our results indicate that rural areas had a larger proportion of older cases (>65 years old) and younger cases (<20 years old) than did the urban areas. It has been reported that rural areas have older populations, on average, and more people with underlying health conditions than suburban and urban communities [14]. Additionally, older adults are more likely to be hospitalized and have severe COVID-19, with higher mortality rates [34]. Household transmission played an important role in both outbreaks, which corroborates previous studies [24,35]. Urban residents are more likely to participate in preventive behaviors than rural residents [36]. In addition, subways, office buildings, and residential garbage are positively connected with the virus transmission [37], which increases the probability of household and social transmission. However, in rural areas, community contacts lead to a large proportion of infections. Similar to other rural areas [38], most community contacts in Hebei are consequent to wedding receptions. Apart from these gathering activities, rural citizens are less likely to seek medical help when feeling sick, and medical professionals are less capable of accurately diagnosing, reporting, and treating cases of infection. Therefore, the government must develop prevention and control measures in rural areas, mainly focusing on older adults and children and restricting large gatherings that pose a high risk for infectious disease transmission [34].

We find that NPIs lead to a larger reduction in infections than vaccination (80% efficacy and 65% coverage). This may explain why the outbreak is swiftly controlled in Hebei. A recent study also verified that NPIs are cost-effective approaches to curb the spread of SARS-CoV-2 [39]. However, the effect of NPIs is closely related to the timing and quality of implementation; hence, similar strategies might have different effects in different cities [40]. Therefore, countries with strong governance can prioritize NPIs until vaccines are widely available. Nonetheless, the durability of responses after vaccination is uncertain [41]. In conclusion, vaccination is more suitable for countries with economic strength but weak governance.

Our findings have several limitations: First, contact tracing of asymptomatic infections is not provided, which may give rise to biased reconstructions of transmission chains. However, detailed records of asymptomatic infections are difficult to collect because they require intensive prospective clinical sampling and screening, which hinders many studies. We imputed this information by assuming missing at random, and our sensitivity analyses can prove the accuracy of our model to a certain extent, but more detailed epidemic data would be helpful. Second, we explicitly set the fixed value of the proportion of asymptomatic infections when estimating the model parameters. Although we conducted sensitivity analyses, improved methodology and more accurate estimation of the asymptomatic proportion is a future research direction. Lastly, the regulation of asymptomatic infections on transmission dynamics needs to be further explored and evidenced.

## 5. Conclusions

Our study has implications for minimizing discrepancy in SARS-CoV-2 transmission between rural and urban areas and control of the global pandemic. The SARS-CoV-2 transmission in both rural and urban areas has strong transmission heterogeneity but is different in terms of age and contact-type distributions. The government should pay equal value to SARS-CoV-2 transmission in rural and urban areas and conduct specific prevention and control measures in rural areas. Older adults and children should receive particular attention in such policies, and community contact should be minimized. Moreover, since NPIs are more effective than vaccinating 65% of the population (80% efficacy), the government must consider a country’s economy and governance when conducting vaccination and NPIs. The rural–urban transmission discrepancy needs to be verified in larger samples, and the contribution of asymptomatic infections to transmission needs to be further explored.

## Figures and Tables

**Figure 1 ijerph-18-05221-f001:**
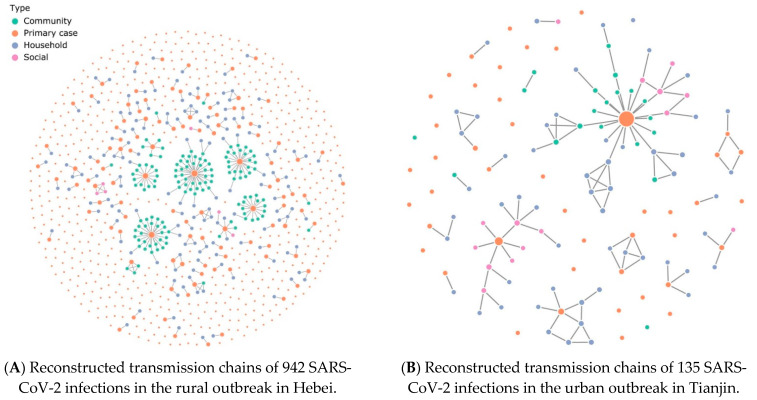
SARS-CoV-2 transmission chains. Each node in the network represents a patient infected with SARS-CoV-2, and each link represents an infector–infectee relationship. The color of the node denotes the reporting contact type of the infected individuals. The size of the node corresponds to the number of secondary cases.

**Figure 2 ijerph-18-05221-f002:**
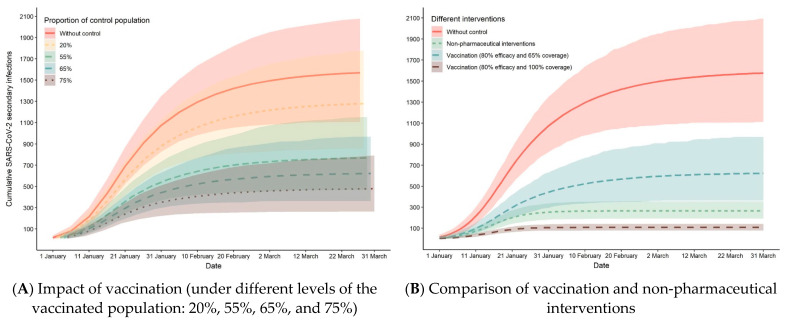
Comparison of NPIs and vaccination in SARS-CoV-2 transmission.

**Table 1 ijerph-18-05221-t001:** Comparison of SARS-CoV-2 transmission between urban and rural areas.

	Urban Area (Tianjin, *n* = 135)	Rural Area (Hebei, *n* = 942)	*p*-Value
Age, years			<0.001
Median (IQR)	49 (36–62)	46 (30–60)	
<20	5 (3.7%)	155 (16.4%)	
20–64	110 (81.4%)	633 (67.2%)	
≥65	20 (14.8%)	152 (16.1%)	
Sex			0.01
Female	63 (46.7%)	553 (58.7%)	
Male	72 (53.3%)	387 (41.1%)	
Contact type			<0.001
Household	55 (61.1%)	147 (51.4%)	
Social	15 (16.7%)	6 (2.1%)	
Community	20 (22.2%)	133 (46.5%)	
Median of serial interval	5.5 (IQR: 3.6–7.8)	6.0 (IQR: 3.6–9.0)	0.73
Transmission dynamics ^†^			
*R*	0.74	0.55	0.16
LRT 95% CI	(0.51, 1.10)	(0.45, 0.68)	
BCa bootstrap 95% CI	(0.53, 3.49)	(0.44, 0.69)	
*k*	0.35	0.14	0.09
LRT 95% CI	(0.13, 1.21)	(0.10, 0.20)	
BCa bootstrap 95% CI	(0.12, 0.95)	(0.10, 0.19)	

^†^ The estimation of *R* and *k* on the basis of the imputed dataset with a total of 1136 infections in the rural area and 135 infections in the urban area (without diagnosed asymptomatic cases in the urban area). Student’s *t*-test was used to compare the differences in age groups. The $\chi^{2}$ test was used to compare differences in sex and contact type. LRT was used to compare the difference in the serial interval and transmission dynamics. IQR, interquartile range.

**Table 2 ijerph-18-05221-t002:** Impact of non-pharmaceutical interventions on SARS-CoV-2 transmission dynamics.

	*R* (95% CI)	*k* (95% CI)
Before first round citywide NAT (<1/09)	0.81 (0.65, 1.02)	0.27 (0.14, 0.56)
During first to second round citywide NAT (1/09–1/14)	0.33 (0.22, 0.50)	0.13 (0.07, 0.23)
After second round citywide NAT (>1/14)	0.36 (0.25, 0.55)	0.17 (0.10, 0.31)

*R*, average reproductive number; CI, confidence interval; *k*, dispersion parameter; NAT, nucleic acid testing; SARS-CoV-2, severe acute respiratory syndrome coronavirus 2.

**Table 3 ijerph-18-05221-t003:** Sensitivity analyses.

	*R*	*k*
Null asymptomatic infections (*p* = 0)	0.51	0.13
20% asymptomatic infections (*p* = 0.2)	0.56	0.14
40% asymptomatic infections (*p* = 0.4)	0.63	0.17
60% asymptomatic infections (*p* = 0.6)	1.62	0.09
80% asymptomatic infections (*p* = 0.8)	1.78	0.10
All asymptomatic infections (*p* = 1)	1.95	0.13
Mean (SD) for 10-fold cross-validation	0.55 (0.02)	0.14 (0.01)

*k*, dispersion parameter; *R*, average reproductive number; SARS-CoV-2, severe acute respiratory syndrome coronavirus 2.

## Data Availability

The code and data to perform statistical analysis is available from https://github.com/githublyy0325/COVIDHeterogeneity.

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
