# Peer review of "Transmission Dynamics, Heterogeneity and Controllability of SARS-CoV-2: A Rural–Urban Comparison"

_ijerph, 2021, doi:10.3390/ijerph18105221_

Round 1
Reviewer 1 Report
Dear Authors,
It is an honor for me to review this excellent manuscript which is really important and relevant with ongoing pandemic. There are some notes to be concerned (I put highlights and notes on the manuscript's file) as it is summarized as below :
- Please add some literature reviews to support the explanation of methodology.
- In supplement document, it will be much better if the Authors can provide all parameters, variables, and notations in a table to make it easier for the readers understanding mathematical models.
- Please recheck the format of references based on MDPI regulation.
Thank you and have a nice day.

Reviewer 2 Report
The present paper has already achieved a sufficient level of completeness therefore the reviewer would like to support the acceptance. However, as minor revisions, the following points can be addressed.
Introduction: There are several papers on COVID-19 infections that have adopted an empirical and/or predictive approach, some of which are relevant to this study. Please add a review of these previous studies.
Section 3.2: 'there were significant differences in age, sex, and contact-type distributions between these two areas', what is the population propensity? Are the differences observed between the given groups the result of simply demographic differences or are they due to characteristics of the route of transmission?
Discussion: 'household contact was the most important mode of transmission, whether in the city or the countryside, but community contact also played an important role in countryside transmission.' What are the factors behind the rural characteristics? Related to 'transmission heterogeneity results from many factors including pathogen virulence, control measures, and activity density,' more elaborate discussion can be expected on the contrast with urban areas.
Conclusions: Only policy recommendations are given, but as an academic paper, a summary of the study and key findings should be provided.
Institutional Review Board Statement: A brief explanation of why it is not applicable should be provided.
Informed Consent Statement: It is still a template thus should be deleted.
Reviewer 3 Report
This article investigates the transmissibility of SARS-CoV-2 in both rural and urban areas with the asymptomatic infections according to some acquired data records from Hebei and Tianjin provinces in China. presents an interesting topic of research. The presented comparative study is very interesting, and it is supported by simulations and analyses. This article provides a good contribution to the existing knowledge in this field of research. This article has some good and interesting results. The discussion of results is also good and explains results in detail. The scope of this paper matches the scope of this prestigious journal, International Journal of Environmental Research and Public Health. However, this paper needs a minor revision in to order to be considered for acceptance in this journal. The following are the recommended suggestions and corrections by the referee:
- The abstract section needs to be revised and re-written briefly. Brief and straight to point abstract will be highly appreciated by all readers.
- This paper suffers from several typos and grammatical errors that need to be carefully corrected.
- Please try to avoid using the past verb in the whole manuscript when you refer to this current study. We use past verb when we refer to previous works in literature review.
- Please try to organize Table 1 because it can cause a confusion to the reader when he/she looks at the Table 1 data.
- Please make sure Table 2 and the table itself in the same page.
- Figure 2 is not clear to the reader. Please insert a high-resolution photo. Another option is to separate both parts (A and B) of Figure 2 where each part is below the other one, so you can increase the size of each Figure 2 part in order to make it clear to the reader.
- If you have applied any particular assumptions in your model or study, please try to explain them briefly in one sentence.
- The conclusions section needs to be improved. Please try to provide some possible future works based on this current research study. In addition, please pot a dot “.” at the end of paragraph in conclusions to stop the paragraph.
In conclusion, editing all those recommended corrections can improve the quality of this research paper.
